# The Concept of “Platinum Sensitivity” in Endometrial Cancer

**DOI:** 10.3390/cancers17152557

**Published:** 2025-08-02

**Authors:** Shoji Nagao, Atsushi Fujikawa, Ryoko Imatani, Yoshinori Tani, Hirofumi Matsuoka, Naoyuki Ida, Junko Haraga, Chikako Ogawa, Keiichiro Nakamura, Hisashi Masuyama

**Affiliations:** Department of Obstetrics and Gynecology, Graduate School of Medicine, Dentistry and Pharmaceutical Sciences, Okayama University, 2-5-1 Shikata-cho, Kita-ku 700-8530, Okayama, Japan; phbi1nz2@okayama-u.ac.jp (A.F.); pryt3xvs@okayama-u.ac.jp (R.I.); pplt247u@okayama-u.ac.jp (Y.T.); ptkp3y1a@okayama-u.ac.jp (H.M.); pnww6nnf@okayama-u.ac.jp (N.I.); jharaga@okayama-u.ac.jp (J.H.); c.fukushima@cc.okayama-u.ac.jp (C.O.); k-nakamu@cc.okayama-u.ac.jp (K.N.); masuyama@cc.okayama-u.ac.jp (H.M.)

**Keywords:** endometrial cancer, platinum sensitivity, platinum free interval

## Abstract

The concept of “platinum sensitivity,” originally established in ovarian cancer, is increasingly recognized as relevant in recurrent endometrial cancer. In both malignancies, a longer platinum-free interval (PFI) is associated with improved efficacy of platinum rechallenge. However, notable differences exist; in ovarian cancer, durable responses extending beyond the PFI are rare (~3%), whereas in recurrent endometrial cancer, approximately 30% of patients exhibit sustained responses beyond this interval. With the recent incorporation of immune checkpoint inhibitors into the therapeutic landscape for endometrial cancer, the optimal integration of the platinum sensitivity concept into clinical decision-making—particularly regarding treatment sequencing and drug selection—remains an important and unresolved challenge. Further research is warranted to elucidate the mechanisms underlying platinum resistance and to inform the development of optimal therapeutic strategies.

## 1. Introduction

Endometrial cancer is the most common gynecologic malignancy in developed countries, with its global incidence steadily rising, largely due to population aging and increasing rates of obesity and metabolic syndrome [1,2]. Although most patients are diagnosed at an early stage and achieve favorable outcomes with surgery alone, a substantial subset of those with advanced or recurrent disease continue to experience poor prognoses [3]. The standard first-line treatment for recurrent or metastatic endometrial cancer has traditionally been platinum-based chemotherapy, most commonly a combination of carboplatin and paclitaxel (TC therapy) [4]. However, therapeutic options after disease progression remain limited, and long-term survival outcomes are generally unsatisfactory [5].

Recent advances in molecular characterization—particularly those from The Cancer Genome Atlas (TCGA)—have enabled the classification of endometrial cancer into distinct four molecular subtypes, each with unique prognostic and therapeutic implications [6]. Additionally, the introduction of immune checkpoint inhibitors has markedly improved outcomes, especially in tumors exhibiting deficient mismatch repair (dMMR) or high microsatellite instability [7,8]. Nonetheless, identifying reliable biomarkers for guiding optimal treatment selection remains an ongoing challenge, and there is currently no consensus regarding the most effective sequencing of therapies.

One emerging concept that may aid therapeutic decision-making is “platinum sensitivity,” a paradigm originally established in ovarian cancer that is now being explored in endometrial cancer to help stratify patients and inform treatment strategies. This review aims to examine the evolving role of platinum sensitivity in endometrial cancer, with a focus on its clinical relevance and potential utility in guiding management decisions.

## 2. The Concept of Platinum Sensitivity in Recurrent Ovarian Cancer

Since the landmark 1991 study by Markman et al. demonstrated that the effectiveness of platinum re-treatment in recurrent ovarian cancer depends on the platinum-free interval (PFI), the concept of platinum sensitivity has become central in guiding treatment selection in this context [9]. “Platinum-free interval”, defined as the duration from the last dose of platinum-based chemotherapy to disease recurrence, is often used to estimate platinum sensitivity. In contrast, “progression-free interval” measures the time from treatment initiation to progression. While both reflect treatment response, the former is particularly relevant in assessing the potential benefit of platinum rechallenge. The correlation between PFI and the efficacy of platinum-based chemotherapy forms the cornerstone of this concept. Patients with recurrent ovarian cancer and a PFI exceeding 6 months are classified as “platinum sensitive” [10,11]. These patients are typically considered eligible for platinum-based second-line chemotherapy, with reported response rates ranging from 27% to 65% and the median overall survival (OS) between 12 and 24 months. For patients with “platinum-sensitive” recurrent ovarian cancer, platinum-based combination regimens such as TC, carboplatin/pegylated liposomal doxorubicin, or carboplatin/gemcitabine, with or without bevacizumab, have shown superior outcomes compared with non-platinum therapies or platinum monotherapy [12]. In contrast, patients with a PFI of less than 6 months are classified as “platinum resistant”. These patients typically have a median OS of 6 to 9 months and exhibit response rates of only 10% to 30% to second-line platinum-based chemotherapy. For this group, non-platinum monotherapy has been shown to be equally effective and less toxic compared to platinum-based combinations.

The introduction of maintenance therapies, including vascular endothelial growth factor inhibitors like bevacizumab and poly ADP-ribose polymerase inhibitors following initial chemotherapy, has introduced ambiguity into the significance of PFI [12]. Although a longer PFI generally correlates with an improved response to platinum rechallenge, the strict dichotomy between “platinum sensitive” and “platinum resistant” is increasingly recognized as arbitrary. Notably, in patients with platinum-resistant recurrent ovarian cancer and a PFI of 3–6 months, platinum-based chemotherapy has demonstrated a superior recurrence-free survival compared with non-platinum-based alternatives [13]. At the 5th Ovarian Cancer Consensus Conference, the PFI paradigm was partially revised, and the concept of the treatment-free interval (TFI) was introduced, considering PFI and incorporating additional factors such as prior bevacizumab exposure, BRCA mutation status, and histological subtype [14]. Reflecting these shifts, the ESMO-ESGO consensus recommendations on ovarian cancer no longer reference PFI [15]. Instead, they recommend re-administering platinum-based agents to patients not deemed ineligible, such as those with recurrence during treatment or who have platinum hypersensitivity.

## 3. The Concept of Platinum Sensitivity in Recurrent Endometrial Cancer

Similarly to ovarian cancer, growing evidence suggests that the concept of platinum sensitivity may also be relevant in endometrial cancer. Moore et al. conducted an ancillary analysis using pooled data from the Gynecologic Oncology Group (GOG) clinical trials to evaluate whether the PFI after platinum-based chemotherapy and TFI after non-platinum-based chemotherapy could predict survival following recurrence in patients with endometrial cancer [16]. This study consisted of two parts: Part 1 included 586 patients from five phase III trials (GOG 107, 122, 139, 163, and 177), while Part 2 included 294 patients from nine phase II salvage therapy trials in the GOG 129 series. In Part 1, a PFI of >6 months was the strongest predictor of improved survival following second-line chemotherapy, associated with a 30% reduction in the risk of death (hazard ratio [HR] 0.70; 95% confidence interval [CI], 0.59–0.84; *P* < 0.0001). In contrast, the re-administration of platinum agents did not significantly improve survival (HR 0.92; 95% CI, 0.77–1.11; *P* = 0.392). In Part 2, a TFI of >3 months was associated with a 25% reduction in the risk of death compared with a TFI of < 3 months (HR 0.75; 95% CI, 0.57–0.97; *P* = 0.030). The authors concluded that the interval between the completion of prior chemotherapy and disease recurrence (PFI or TFI) may have served as a prognostic indicator for OS following subsequent chemotherapy in patients with advanced or recurrent endometrial cancer, although no evidence was found to support their utility in guiding therapeutic decision-making.

The Sankai Gynecologic Study Group (SGSG) and the Gynecologic Oncology Trial and Investigation Consortium (GOTIC) conducted a retrospective study involving 262 patients from 30 institutions in Japan who received platinum-based chemotherapy as initial treatment and were re-treated with platinum agents upon recurrence (SGSG-012/GOTIC-004/Intergroup study). The study assessed the association between the PFI and treatment outcomes, including the response rate to second-line therapy, progression-free survival (PFS), and OS [17]. Longer PFIs were significantly correlated with improved treatment outcomes. Specifically, the response rates to second-line platinum therapy increased with longer PFIs, ranging from 25% in patients with a PFI of < 6 months to 65% in those with a PFI of ≥ 24 months (Table 1). The median PFS and OS also improved progressively with increasing PFIs, with markedly better outcomes observed in patients with a PFI of ≥12 months compared with those with shorter intervals (10.3 vs. 4.4 months and 40.9 vs. 13.8 months, respectively; *p* < 0.0001). Furthermore, this study found that a combination of carboplatin and a taxane (either paclitaxel or docetaxel) was more effective than adriamycin plus cisplatin as a second-line chemotherapy regimen for recurrent endometrial cancer [18]. These findings provide robust evidence that the PFI may serve not only as a prognostic factor but also as a predictive marker of responses to platinum re-treatment.

The concept of platinum sensitivity is applicable to endometrial cancer as well as ovarian cancer. Thus, in patients with recurrent endometrial cancer and a sufficiently long PFI, the re-administration of platinum-based therapy represents a valid treatment approach. A longer PFI has been consistently associated with an improved prognosis in recurrent endometrial cancer, including prolonged PFS and OS. As demonstrated in Table 1, the response rates increase in parallel with longer PFIs, supporting its utility not only as a prognostic indicator but also as a predictor of treatment efficacy. However, the efficacy of platinum rechallenge is linearly correlated with PFI duration, and the distinction between platinum sensitivity and resistance remains inherently ambiguous and inevitably arbitrary. Furthermore, despite PFIs being clinically relevant as a prognostic and predictive marker, it has not been incorporated into major guidelines such as those of the National Comprehensive Cancer Network (NCCN) or European Society of Gynaecological Oncology (ESGO), due to the lack of prospective validation in endometrial cancer. Its current use is based on retrospective data and should be interpreted with caution in clinical practice.

## 4. Another Aspect of the Concept of Platinum Sensitivity

Markman et al. evaluated chemotherapy outcomes in 176 patients with recurrent ovarian cancer treated at the Cleveland Clinic and found that the duration of response to initial platinum-containing chemotherapy strongly influenced the maximum achievable duration of response to platinum-based therapy upon relapse [19]. Notably, only four (3%) of the assessable patients experienced a duration of secondary platinum response (DSPR) that exceeded their preceding PFI (Table 2). Although most second-line regimens included agents not used in prior treatments, patients whose initial response to platinum-based therapy lasted ≥18 months demonstrated a 75% response rate to subsequent chemotherapy, compared with only 33% in those whose initial response was <12 months. Thus, the concept of platinum sensitivity in ovarian cancer encompasses both a favorable prognostic aspect—namely that the efficacy of platinum rechallenge improves with longer PFIs—and a key limitation: the DSPR rarely exceeds the preceding PFI.

Our clinical research group conducted an ancillary analysis of the SGSG-012/GOTIC-004/Intergroup study to determine whether a similar limitation applies to recurrent endometrial cancer [20]. Among 279 patients who received second-line platinum-based chemotherapy, 130 (47%) achieved an objective treatment response. In contrast to recurrent ovarian cancer, a notable finding was that 28 of 48 patients (58%) with a PFI of <12 months and 10 of 43 patients (23%) with a PFI between 12 and 24 months—comprising 40 of the 130 responders (31%)—experienced a DSPR that exceeded their prior PFI (Table 2). Furthermore, 51 patients (38%) achieved a DSPR longer than 12 months, and 8 patients (6%), including 3 with a PFI of <12 months, achieved a DSPR exceeding 36 months. These findings suggest a divergent pattern in endometrial cancer, wherein the PFI does not strictly determine DSPR, unlike the trend observed in ovarian cancer.

In recurrent endometrial cancer, as in recurrent ovarian cancer, longer PFIs are associated with higher response rates to platinum rechallenge, as well as prolonged PFS and OS, all demonstrating linear correlations. These findings suggest that the concept of platinum sensitivity is applicable to recurrent endometrial cancer and may serve as a useful framework for both prognostic assessment and treatment decision-making. However, the threshold separating “platinum-sensitive” from “platinum-resistant” disease remains inherently ambiguous and lacks a precise definition. Importantly, unlike recurrent ovarian cancer—where the DSPR is tightly constrained by the preceding PFI—recurrent endometrial cancer does not exhibit such a strict relationship. In a substantial subset of patients, the DSPR exceeded the prior PFI. This observation implies that the biological mechanisms underlying the development or reversal of platinum resistance may differ between endometrial and ovarian cancers. Platinum resistance in ovarian cancer is widely understood to be multifactorial. Key mechanisms include alterations in DNA repair pathways, particularly deficiencies in homologous-recombination repair involving *BRCA* mutations and their reversions [21,22]. Reduced intracellular accumulation of platinum due to changes in drug transport has also been identified as a contributing factor [23]. Additional repair processes, such as mismatch repair deficiency and replication fork protection, have been implicated in resistance development [24]. The tumor microenvironment and immune regulatory pathways, including PD-1/PD-L1 signaling, are increasingly recognized as playing roles in mediating resistance [25]. Epigenetic modifications are also believed to contribute to resistant phenotypes [26]. In contrast, platinum resistance in endometrial cancer remains largely unexplored, and no common mechanisms with ovarian cancer have been clearly established. Further investigation into “super-responders,” patients who demonstrate exceptionally prolonged responses to platinum rechallenge, may offer valuable insights into the molecular determinants of this differential platinum responsiveness.

## 5. Platinum Sensitivity in the Context of Recent Treatment Strategies

Immune checkpoint inhibitors (ICIs) have recently been incorporated into the treatment of endometrial cancer, either as a monotherapy or in combination with tyrosine kinase inhibitors such as lenvatinib or platinum-based chemotherapy, such as the TC regimen [27,28,29,30,31,32]. With the introduction of ICIs, treatment selection for recurrent endometrial cancer now requires consideration not only of platinum sensitivity but also of mismatch repair (MMR) status. Beyond its role in selecting candidates for ICIs, molecular classification has also been linked to differential sensitivity to platinum and taxane chemotherapy [33]. *p53*-abnormal tumors appear to benefit most from combined chemoradiotherapy, as shown in the PORTEC-3 trial. In contrast, *POLE*-mutated tumors show excellent prognosis regardless of chemotherapy, supporting de-escalation strategies. Microsatellite instability-high (MSI-H)/dMMR tumors are typically treated with platinum-based regimens initially, though immune checkpoint inhibitors are preferred upon recurrence. NSMP tumors show variable responses to chemotherapy, reflecting their biological heterogeneity. These findings suggest that molecular subtypes may help refine chemotherapy decisions in endometrial cancer.

The phase III ENGOT-en9/LEAP-001 trial directly compared the efficacy and safety of lenvatinib plus pembrolizumab (LEN + PEM) with TC therapy in patients with stage III/IV or recurrent endometrial cancer [34]. A total of 842 patients, including 642 patients with proficient mismatch repair (pMMR) tumors, were randomized to receive either lenvatinib plus pembrolizumab or paclitaxel plus carboplatin. In the overall population, LEN + PEM demonstrated numerically favorable but statistically non-significant improvements in PFS [12.5 vs. 10.2 months; HR 0.91 (95%CI: 0.76–1.09)] and OS [37.7 vs. 32.1 months; HR 0.93 (95%CI: 0.77–1.12)]. In the pMMR subgroup, LEN + PEM did not yield statistically significant improvements in either PFS [9.6 vs. 10.2 months; HR 0.99 (95%CI: 0.82–1.21)] or OS [30.9 vs. 29.4 months; HR 1.02 (95%CI: 0.83–1.26)]. In contrast, exploratory analyses revealed that patients with dMMR tumors derived greater benefits from LEN + PEM, with HRs of 0.61 (95%CI: 0.40–0.92) for PFS and 0.57 for OS (95%CI: 0.36–0.91). Similarly, patients previously treated with (neo)adjuvant chemotherapy appeared to benefit more from LEN + PEM than from standard chemotherapy, with HRs of 0.52 (95%CI: 0.33–0.82) for PFS and 0.64 (95%CI: 0.40–1.03) for OS. These findings suggest that LEN + PEM may be more effective than TC therapy in dMMR tumors. In contrast, in pMMR tumors, no definitive difference in efficacy was observed between the two regimens.

Table 3 summarizes major clinical trials that have evaluated ICI-containing regimens in patients with endometrial cancer [28,29,30,31,32,34]. Although these five randomized controlled trials are not directly comparable, TC therapy was consistently employed in the control arms across all studies. Accordingly, the HRs of the investigational arms are presented in parallel to provide an overview of the general trend. Two key trends can be observed from Table 3. First, in all trials, the HRs were consistently lower in patients with dMMR compared with those with pMMR, indicating that the added benefit of ICIs to TC therapy—whether as a combination or maintenance treatment—is greater in dMMR tumors. This pattern suggests that the efficacy of ICIs is influenced by MMR status. Second, no consistent trend is observed regarding the effect of prior chemotherapy exposure on the effectiveness of ICIs. Additionally, a subset analysis of the KEYNOTE-775 trial—the pivotal study leading to the development of LEN + PEM—demonstrated that the efficacy of LEN + PEM is not influenced by the PFI [34].

As discussed above, while the efficacy of TC therapy appears to be influenced by the PFI, the therapeutic activity of ICIs and lenvatinib is considered to be independent of the PFI. Theoretically, when choosing between TC + ICI and LEN + PEM as a treatment regimen, the PFI at which the relative efficacy of TC and lenvatinib reverses may serve as a reference point for regimen selection. In patients with recurrent endometrial cancer and a relatively long PFI, TC therapy is presumed to be more effective than lenvatinib, suggesting that a TC + ICI regimen may be the more appropriate treatment option. Conversely, in patients with a short PFI, lenvatinib is expected to be more effective than TC therapy, making the LEN + PEM regimen a more appropriate treatment option. 

Based on the PFI and MMR status, the optimal treatment strategy can be categorized as shown in Table 4. In patients with recurrent endometrial cancer and a PFI of ≤6 months, the efficacy of TC therapy is limited. However, subset analyses of the KEYNOTE-775 trial demonstrated that LEN + PEM is more effective than non-platinum-based regimens regardless of MMR status, suggesting that LEN + PEM may be the preferred option in this setting [34]. In patients with a PFI of ≥12 months, TC therapy is expected to be effective, and the addition of ICIs may further enhance treatment outcomes. Therefore, TC + ICI is considered the preferred option in this subgroup. However, in patients with pMMR tumors, the addition of ICI has not been shown to improve OS (Table 3), suggesting that TC monotherapy may still be a viable treatment option for patients with pMMR. For patients in the intermediate category with a PFI of 6 to 12 months, the relative efficacy of TC and lenvatinib is likely to shift within this range, as the efficacy of TC rechallenge is linearly correlated with PFI duration. Therefore, determining the optimal treatment between TC + ICI and LEN + PEM may require additional molecular, genetic, or clinical biomarkers. 

## 6. Conclusions

The concept of “platinum sensitivity”, long established in ovarian cancer, may also be applicable to recurrent endometrial cancer, providing a useful framework for prognostic assessment and guiding therapeutic decision-making. Despite the recent integration of ICIs into the treatment landscape for endometrial cancer, the clinical relevance of platinum sensitivity remains intact. However, further accumulation of molecular, genetic, or clinical evidence is required to enable more precise decision-making for patients with recurrent endometrial cancer. To further elucidate the clinical relevance of PFI in recurrent endometrial cancer, it is essential to assess its utility within each molecular subtype and to perform a molecular characterization of so-called “super-responders”.

## Figures and Tables

**Table 1 cancers-17-02557-t001:** Correlation between platinum-free interval and efficacy and prognosis during platinum.

re-administration [17]				
	Platinum-free interval (months)			
	PFI < 6	6 ≤ PFI < 12	12 ≤ PFI < 24	24 ≤ PFI
ORR (%)	25	38	62	65
mPFS (95%CI) (months)	3.2 (2.3–4.3)	6.0 (4.4–7.3)	7.8 (5.8–10.6)	13.4 (10.2–20.0)
	4.4 (3.7–5.8) *		10.3 (8.2–12.6) *	
mOS (95%CI) (months)	11.3 (7.9–17.5)	14.8 (11.5–19.5)	27.8 (16.6–∞)	43.0 (27.4–74.7)
	13.8 (10.6–18.1) **		40.9 (25.3–54.2) **	

PFI: platinum-free interval, ORR: overall response rate, mPFS: median progression-free survival, mOS: median overall survival. * Estimates of progression-free survival for patients with platinum-free intervals of <12 months and ≥12 months (matched patients). ** Estimates of overall survival for patients with platinum-free intervals of <12 months and ≥12 months (matched patients).

**Table 2 cancers-17-02557-t002:** Relationship between duration of secondary platinum response and platinum-free interval [19,20].

PFI	DSPR	Ovarian cancer		Endometrial cancer		
(months)	(months)	N	PFI < DSPR	N	PFI < DSPR	
<12	<12	16	1	34	14	
	12≤, <24	0	0	8	8	
	24≤, <36	0	0	3	3	
	36≤	0	0 (6%)	3	3	(58%)
	<12	40	3	25	0	
	12≤, <24	12	0	13	5	
	24≤, <36	0	0	3	3	
	36≤	0	0 (6%)	2	2	(23%)
24≤, <36	<12	15	0	11	0	
	12≤, <24	11	0	5	0	
	24≤, <36	1	0	3	1	
	36≤	0	0 (0%)	0	0	(5%)
36≤	<12	6	0	9	0	
	12≤, <24	8	0	5	0	
	24≤, <36	5	0	3	0	
	36≤	3	0 (0%)	3	1	(5%)
Total		117	4 (3%)	130	40	(31%)

PFI: platinum-free interval, DSPR: duration of secondary platinum response.

**Table 3 cancers-17-02557-t003:** Background and hazard ratios for TC therapy among major clinical trials related to immune check point inhibitors in endometrial cancer.

Study			NRG GY-018	RUBY [30,31]	AtTEnd [32]	DUO-E [29]		LEAP-001[34]
			KN868 [28]					
Drugs			TC	TC	TC	TC	TC/Olaparib	Lenvatinib
			Pembrolizumab	Dostarlimab	Atezolizumab	Durvalumab	Durvalumab	Pembrolizumab
N			819	494	550	718		842
Eligibility		stage	III/IV	III/IV	III/IV	III/IV		III/IV
		PFI	≥12 m	≥6 m	≥6 m	≥12 m		≥6 m
		carcinosarcoma	0%	10%	0%	7%		0%
		non-endometrioid	20%	45%	36%	41%		33%
Duration			14 cycles	3 years	until PD	until PD		until PD
Primary endpoint			PFS	PFS, OS	PFS, OS	PFS		PFS, OS
PFS	overall	HR		0.64	0.74	0.71	0.55	0.91
		(95%CI)		(0.51–0.80)	(0.61–0.91)	(0.57–0.89)	(0.43–0.69)	(0.76–1.09)
	dMMR	HR	0.45	0.28	0.36	0.42	0.41	0.61
		(95%CI)	(0.27–0.73)	(0.16–0.50)	(0.23–0.57)	(0.22–0.80)	(0.21–0.75)	(0.40–0.92)
	pMMR	HR	0.64	0.76	0.92	0.77	0.57	0.99
		(95%CI)	(0.49–0.85)	(0.59–0.98)	(0.73–1.16)	(0.80–0.97)	(0.44–0.73)	(0.82–1.21)
	overall	HR		0.69	0.82	0.77	0.59	0.93
OS		(95%CI)		(0.54–0.89)	(0.63–1.07)	(0.56–1.07)	(0.42–0.83)	(0.77–1.12)
	dMMR	HR	0.55	0.32	0.41	0.34	0.28	0.57
		(95%CI)	(0.25–1.19)	(0.17–0.63)	(0.22–0.76)	(0.13–0.79)	(0.10–0.68)	(0.36–0.91)
	pMMR	HR	0.79	0.79	1.00	0.91	0.69	1.02
		(95%CI)	(0.53–1.17)	(0.60–1.04)	(0.74–1.35)	(0.64–1.30)	(0.47–1.00)	(0.83–1.26)
Previous chemotherapy N			162	NR	129	156		119
	overall	HR				1.15	0.55	0.52
PFS		(95%CI)				(0.74–1.79)	(0.35–0.86)	(0.33–0.82)
	pMMR	HR	0.80		0.68	1.15	0.59	0.60
		(95%CI)	(1.50–1.27)		(0.47–0.97)	(0.73–1.80)	(0.37–0.94)	(0.37–0.97)
	overall	HR						0.64
OS		(95%CI)						(0.40–1.03)
	pMMR	HR						0.67
		(95%CI)						(0.41–1.11)

TC: paclitaxel + carboplatin, PFI: platinum-free interval, PD: disease progression, PFS: progression-free survival, OS: overall survival, HR: hazard ratio, dMMR: deficient mismatch repair, pMMR: proficient mismatch repair.

**Table 4 cancers-17-02557-t004:** The preferred therapeutic regimen.

	PFI		
	<6 m	6 m≤, >12 m	12 m≤
dMMR	LEN + PEM	LEN + PEM or TC + ICI	TC + ICI
pMMR	LEN + PEM	LEN + PEM or TC + ICI	TC + ICI or TC

PFI: platinum-free interval, dMMR: deficient mismatch repair, pMMR: proficient mismatch repair, LEN: lenvatinib, PEM: pembrolizumab, TC: paclitaxel + carboplatin, ICI: immune-checkpoint inhibitor.

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
