# Peer review of "The Concept of “Platinum Sensitivity” in Endometrial Cancer"

_cancers, 2025, doi:10.3390/cancers17152557_

Round 1

Reviewer 1 Report

Comments and Suggestions for Authors

Nagao et al. have written an informative review on the concept of »platinum sensitivity« in endometrial cancer. They describe a series of clinical trials that provide important information about the response to platinum drugs in combination with taxanes in patients with ovarian cancer and endometrial cancer, and they compare platinum therapy with recently introduced targeted therapies, including kinase inhibitors and immune checkpoint inhibitors in patients with endometrial cancer. The authors conclude that »the concept of platinum sensitivity may be applicable in recurrent endometrial cancer«, but that further evidence is needed for precise invidualised drug selection.

The authors explain the concept of platinum sensitivity in ovarian cancer and use a prolonged platinum-free interval (PFI) as a criterion. In the area of chemosensitivity/chemoresistance, also the criterion of progression-free interval is often used, so it would be helpful for readers if an explanation of both terms were added.

The molecular classification of endometrial cancer has contributed to the selection of patients who respond to immune checkpoint inhibitors. Are there any data on sensitivity to treatment with platinum/taxanes in different molecular subtypes of endometrial cancer?

The authors state that »the biological mechanisms underlying the development or reversal of platinum resistance may differ between endometrial and ovarian cancers«." The authors should briefly explain the state of knowledge on the mechanisms of chemoresistance in ovarian and endometrial cancer.

It is advisable to conclude the review with suggestions for further clinical trials that would contribute to optimised treatment of patients with recurrent endometrial cancer.

Minor comment:

The following titles should be corrected:

Table 1. Correlation between platinum-free interval and efficacy and prognosis during platinum
re-administration17)

Table 2. Relationship between duration of secondary platinum response and platinum-free interval19)20).

Author Response

Comment 1

The authors explain the concept of platinum sensitivity in ovarian cancer and use a prolonged platinum-free interval (PFI) as a criterion. In the area of chemosensitivity/chemoresistance, also the criterion of progression-free interval is often used, so it would be helpful for readers if an explanation of both terms were added.

Response: Thank you for your suggestion. To improve clarity for readers, we have added a concise definition and differentiation between platinum-free interval and progression-free interval in the 2nd section.

The added explanation is as follows: "Platinum-free interval", defined as the duration from the last dose of platinum-based chemotherapy to disease recurrence, is often used to estimate platinum sensitivity. In contrast, "progression-free interval" measures the time from treatment initiation to progression. While both reflect treatment response, the former is particularly relevant in assessing the potential benefit of platinum rechallenge.

Comment 2

The molecular classification of endometrial cancer has contributed to the selection of patients who respond to immune checkpoint inhibitors. Are there any data on sensitivity to treatment with platinum/taxanes in different molecular subtypes of endometrial cancer?

Response: We appreciate this important question. We have now added discussion on this topic in the 5th section. While immune checkpoint inhibitors show differential efficacy by molecular subtype, data on platinum sensitivity stratified by TCGA classification remain limited.

The added explanation is as follows: Beyond its role in selecting candidates for ICIs, molecular classification has also been linked to differential sensitivity to platinum and taxane chemotherapy. p53-abnormal tumors appear to benefit most from combined chemoradiotherapy, as shown in the PORTEC-3 trial. In contrast, POLE-mutated tumors show excellent prognosis regardless of chemotherapy, supporting de-escalation strategies. MSI-H/dMMR tumors are typically treated with platinum-based regimens initially, though immune checkpoint inhibitors are preferred upon recurrence. NSMP tumors show variable response to chemotherapy, reflecting their biological heterogeneity. These findings suggest that molecular subtypes may help refine chemotherapy decisions in endometrial cancer.

Comment 3

The authors state that the biological mechanisms underlying the development or reversal of platinum resistance may differ between endometrial and ovarian cancers. The authors should briefly explain the state of knowledge on the mechanisms of chemoresistance in ovarian and endometrial cancer.

Response: Thank you. We have included a new paragraph summarizing the current knowledge of platinum resistance mechanisms in both malignancies. We added the discussion in the 4th section.

The added discussions are as follows: Platinum resistance in ovarian cancer is widely understood to be multifactorial. Key mechanisms include alterations in DNA repair pathways, particularly deficiencies in homologous recombination repair involving BRCA mutations and their reversions.Reduced intracellular accumulation of platinum due to changes in drug transport has also been identified as a contributing factor. Additional repair processes, such as mismatch repair deficiency and replication fork protection, have been implicated in resistance development. The tumor microenvironment and immune regulatory pathways, including PD-1/PD-L1 signaling, are increasingly recognized as playing roles in mediating resistance. Epigenetic modifications are also believed to contribute to resistant phenotypes. In contrast, platinum resistance in endometrial cancer remains largely unexplored, and no common mechanisms with ovarian cancer have been clearly established.

Comment 4

It is advisable to conclude the review with suggestions for further clinical trials that would contribute to optimised treatment of patients with recurrent endometrial cancer.

Response: Thank you for this suggestion. We have expanded our conclusion to emphasize future research directions.

The added sentence is as follows: To further elucidate the clinical relevance of PFI in recurrent endometrial cancer, it is essential to assess its utility within each molecular subtype and to perform molecular characterization of so-called super-responders.

Minor comment:

The following titles should be corrected:

Table 1. Correlation between platinum-free interval and efficacy and prognosis during platinum
re-administration17)

Table 2. Relationship between duration of secondary platinum response and platinum-free interval19)20).

Response: Thank you. Citations in Table 1 and Table 2 have been standardized.

Reviewer 2 Report

Comments and Suggestions for Authors

The manuscript is clearly written, comprehensive in its literature coverage, and supported by well-organized tables that enhance the reader’s understanding.

One of the main strengths of this article is the discussion of immunotherapy and antiangiogenic agents reshaping the treatment landscape, the question of when and how to reintroduce platinum-based chemotherapy in endometrial cancer is increasingly pertinent. Still, there are several areas where the manuscript could be strengthened:

-First, the role of PFI as a prognostic versus predictive marker could be clarified. While it is evident from the text that PFI correlates with survival, the evidence for its ability to predict benefit from platinum rechallenge remains inconclusive. A more explicit distinction between these concepts would improve interpretability for readers.

-the manuscript would benefit from a brief discussion of the practical limitations in applying PFI as a clinical decision tool. Since this marker is not formally included in treatment guidelines and lacks prospective validation in endometrial cancer, acknowledging this limitation would lend further balance to the narrative.

-Briefly expand on the biological rationale behind the observed differences in PFI and DSPR between endometrial and ovarian cancers. While the authors allude to possible biological distinctions, a short paragraph on molecular mechanisms (such as differences in tumor microenvironment or hormonal regulation) could enrich the discussion.

-A small typographical correction is needed where “renew and editing” should be corrected to “review and editing.” Additionally, terms such as “pMMR” (proficient mismatch repair) should be defined consistently, as they appear multiple times across the manuscript.

Author Response

Response to reviewer

We would like to express our sincere appreciation to the Editor and the Reviewers for their valuable time and thoughtful feedback. Their comments have been highly constructive and have helped us to improve the manuscript. We have revised the manuscript accordingly and provided detailed responses to each comment. Below, we provide a point-by-point response to each of the reviewers’ comments. All changes made to the manuscript are highlighted in red in the revised version.

Reviewer#2

Comment 1

First, the role of PFI as a prognostic versus predictive marker could be clarified. While it is evident from the text that PFI correlates with survival, the evidence for its ability to predict benefit from platinum rechallenge remains inconclusive. A more explicit distinction between these concepts would improve interpretability for readers.

Response: Thank you. We agree that this distinction is critically important for accurate clinical interpretation. We have further clarified this point in the 3rd section to emphasize the current evidence base. Based on the results of the SGSG-012/GOTIC-004/Intergroup study, we believe that the PFI is useful not only for prognostic prediction but also as an efficacy prediction tool.

The added discussions are as follows: A longer PFI has been consistently associated with improved prognosis in recurrent endometrial cancer, including prolonged PFS and OS. As demonstrated in Table 1, response rates increase in parallel with longer PFIs, supporting its utility not only as a prognostic indicator but also as a predictor of treatment efficacy.

Comment 2

The manuscript would benefit from a brief discussion of the practical limitations in applying PFI as a clinical decision tool. Since this marker is not formally included in treatment guidelines and lacks prospective validation in endometrial cancer, acknowledging this limitation would lend further balance to the narrative.

Response: Thank you. We have added this important limitation to the revised manuscript on 3rd section.

The added discussions regarding limitation of PFI are as follows: Furthermore, despite PFI is clinically relevant as a prognostic and predictive marker, it has not been incorporated into major guidelines such as those of the NCCN or ESGO, due to the lack of prospective validation in endometrial cancer. Its current use is based on retrospective data and should be interpreted with caution in clinical practice.

Comment 3

Briefly expand on the biological rationale behind the observed differences in PFI and DSPR between endometrial and ovarian cancers. While the authors allude to possible biological distinctions, a short paragraph on molecular mechanisms (such as differences in tumor microenvironment or hormonal regulation) could enrich the discussion.

Response: Thank you. We have included a new paragraph summarizing the current knowledge of platinum resistance mechanisms in both malignancies. We added the discussion in the 4th section.

The added discussions are as follows: Platinum resistance in ovarian cancer is widely understood to be multifactorial. Key mechanisms include alterations in DNA repair pathways, particularly deficiencies in homologous recombination repair involving BRCA mutations and their reversions.Reduced intracellular accumulation of platinum due to changes in drug transport has also been identified as a contributing factor. Additional repair processes, such as mismatch repair deficiency and replication fork protection, have been implicated in resistance development. The tumor microenvironment and immune regulatory pathways, including PD-1/PD-L1 signaling, are increasingly recognized as playing roles in mediating resistance. Epigenetic modifications are also believed to contribute to resistant phenotypes. In contrast, platinum resistance in endometrial cancer remains largely unexplored, and no common mechanisms with ovarian cancer have been clearly established.

Comment 4

A small typographical correction is needed where “renew and editing” should be corrected to “review and editing.” Additionally, terms such as “pMMR” (proficient mismatch repair) should be defined consistently, as they appear multiple times across the manuscript.

Response: Thank you for your meticulous attention to detail. We have thoroughly reviewed the manuscript and corrected all mentioned typographical errors. "Renew and editing" has been changed to "review and editing," and definitions of "pMMR" (proficient mismatch repair) and "dMMR" (deficient mismatch repair) have been clarified and used consistently at their first mention and in the abbreviations section.

Reviewer 3 Report

Comments and Suggestions for Authors

Dear Authors,

Thank you for the opportunity to review your manuscript, "The Concept of 'Platinum Sensitivity' in Endometrial Cancer". This is a well-structured and timely review that addresses the important issue of adapting the concept of platinum sensitivity to the recent advances in immunotherapy for endometrial cancer.

The paper summarizes the existing data, draws interesting parallels with ovarian cancer, and proposes a practical classification for clinical use. To improve the scientific accuracy of the manuscript, I recommend the following minor revisions:

  1. In Table 2 (page 4), the study by Markman et al. is reported to include 117 patients. However, the original article states that data for both PFI and DSPR were available for 121 patients, and this number was used to calculate the percentage of patients with DSPR>PFI (4 out of 121, or 3%). Please clarify the source of the N=117 value.
  2. Table 3 (page 6) contains several discrepancies with the most recent publications:
  3. A) NRG-GY018: The reported PFS values for the dMMR (HR 0.34) and pMMR (HR 0.57) subgroups do not align with the final publication (Eskander et al., 2025; PMID: 40044930).
  4. B) RUBY: An updated analysis is available (Powell M.A. et al., Annals of Oncology, 2024; https://www.annalsofoncology.org/article/S0923-7534(24)00721-X/fulltext).

I recommend that the authors verify and update the HR values and other metrics in this table to ensure they are consistent with the latest published data.

Minor editorial corrections:

  1. The citation format in the caption for Table 2 (page 4) is non-standard: (19)20)).
  2. In the note to Table 3 (page 6), there is a typographical error in "mismatch": “mismach repair-deficient”.
  3. In the caption for Table 4 (page 7), there is a typographical error in "platinum": “platibum free interval”.
  4. On page 7, line 252, there is a typographical error: “writing-renew and editing” should be “writing-review and editing”.

Author Response

Response to reviewer

We would like to express our sincere appreciation to the Editor and the Reviewers for their valuable time and thoughtful feedback. Their comments have been highly constructive and have helped us to improve the manuscript. We have revised the manuscript accordingly and provided detailed responses to each comment. Below, we provide a point-by-point response to each of the reviewers’ comments. All changes made to the manuscript are highlighted in red in the revised version.

Reviewer#3

Comments

  1. In Table 2 (page 4), the study by Markman et al. is reported to include 117 patients. However, the original article states that data for both PFI and DSPR were available for 121 patients, and this number was used to calculate the percentage of patients with DSPR>PFI (4 out of 121, or 3%). Please clarify the source of the N=117 value.

Response: The total number of cases was derived from Table 3 in the study by Markman et al. (J Clin Oncol. 2004;22:3123) [Reference 19], in which the sum of all cases amounted to 117.

  1. Table 3 (page 6) contains several discrepancies with the most recent publications:
  2. A) NRG-GY018: The reported PFS values for the dMMR (HR 0.34) and pMMR (HR 0.57) subgroups do not align with the final publication (Eskander et al., 2025; PMID: 40044930).
  3. B) RUBY: An updated analysis is available (Powell M.A. et al., Annals of Oncology, 2024; https://www.annalsofoncology.org/article/S0923-7534(24)00721-X/fulltext).

I recommend that the authors verify and update the HR values and other metrics in this table to ensure they are consistent with the latest published data.

Response: Thank you very much for your valuable comments. We have updated Table 3 based on the final publication of NRG-GY018 (Eskander et al., NEJM 2025, PMID: 40044930) and the updated RUBY analysis (Powell et al., Ann Oncol 2024). All hazard ratios and subgroup data have been corrected and verified.

Minor editorial corrections:

  1. The citation format in the caption for Table 2 (page 4) is non-standard: (19)20)).
  2. In the note to Table 3 (page 6), there is a typographical error in "mismatch": “mismach repair-deficient”.
  3. In the caption for Table 4 (page 7), there is a typographical error in "platinum": “platibum free interval”.
  4. On page 7, line 252, there is a typographical error: “writing-renew and editing” should be “writing-review and editing”.

Response: In response to your suggestions, we have made the corresponding revisions.